# Investigation of the Antifungal and Anti-Aflatoxigenic Potential of Plant-Based Essential Oils against *Aspergillus flavus* in Peanuts

**DOI:** 10.3390/jof6040383

**Published:** 2020-12-21

**Authors:** Premila Narayana Achar, Pham Quyen, Emmanuel C. Adukwu, Abhishek Sharma, Huggins Zephaniah Msimanga, Hanumanthu Nagaraja, Marikunte Yanjarappa Sreenivasa

**Affiliations:** 1Department of Molecular and Cellular Biology, Kennesaw State University, Kennesaw, GA 30144, USA; qtpham6274@gmail.com; 2Centre for Research in Biosciences, University of the West of England, Bristol BS16 1QY, UK; Emmanuel.Adukwu@uwe.ac.uk; 3Amity Food and Agriculture Foundation, Amity University Uttar Pradesh, Noida 201313, India; asharma5@amity.edu; 4Department of Chemistry and Biochemistry, Kennesaw State University, Kennesaw, GA 30144, USA; hmsimang@kennesaw.edu; 5Department of Studies in Microbiology, University of Mysore, Mysore 570006, India; nagarajh82@gmail.com (H.N.); mys@microbiology.uni-mysore.ac.in (M.Y.S.)

**Keywords:** *Aspergillus flavus*, peanuts, essential oils, antifungal activity, aflatoxin

## Abstract

*Aspergillus* species are known to cause damage to food crops and are associated with opportunistic infections in humans. In the United States, significant losses have been reported in peanut production due to contamination caused by the *Aspergillus* species. This study evaluated the antifungal effect and anti-aflatoxin activity of selected plant-based essential oils (EOs) against *Aspergillus flavus* in contaminated peanuts, Tifguard, runner type variety. All fifteen essential oils, tested by the poisoned food technique, inhibited the growth of *A. flavus* at concentrations ranging between 125 and 4000 ppm. The most effective oils with total clearance of the *A. flavus* on agar were clove (500 ppm), thyme (1000 ppm), lemongrass, and cinnamon (2000 ppm) EOs. The gas chromatography-mass spectrometry (GC-MS) analysis of clove EO revealed eugenol (83.25%) as a major bioactive constituent. An electron microscopy study revealed that clove EO at 500 ppm caused noticeable morphological and ultrastructural alterations of the somatic and reproductive structures. Using both the ammonia vapor (AV) and coconut milk agar (CMA) methods, we not only detected the presence of an aflatoxigenic form of *A. flavus* in our contaminated peanuts, but we also observed that aflatoxin production was inhibited by clove EO at concentrations between 500 and 2000 ppm. In addition, we established a correlation between the concentration of clove EO and AFB1 production by reverse-phase high-performance liquid chromatography (HPLC). We demonstrate in our study that clove oil could be a promising natural fungicide for an effective bio-control, non-toxic bio-preservative, and an eco-friendly alternative to synthetic additives against *A. flavus* in Georgia peanuts.

## 1. Introduction

Contamination of food commodities by toxigenic fungi and the presence of mycotoxins during pre-harvest and post-harvest has attracted the attention of scientific, political, and economic organizations. Mycotoxins are toxins produced naturally by several types of molds [1]. Among the group of mycotoxins (aflatoxins B1, B2, G1, and G2) produced by *Aspergillus flavus* and *Aspergillus parasiticus,* aflatoxin B1 (AFB1) has been classified as a class I human carcinogen by the International Agency for Research on Cancer and is reportedly the most toxic [2,3]. The aflatoxins produced by these two species are known to affect crops such as peanut, maize, yams, cassava, and cereals, recognized as basic staple diets globally, particularly in Africa [4]. It is estimated that 25% or more of global food crops are destroyed annually due to aflatoxins [5]. These economic losses are suffered at a global level with significant public health consequences, and it is estimated that losses due to aflatoxin contamination in the corn industry are between USD 52.1 million and USD 1.68 billion annually [6].

More specifically, AFB1 production is a serious problem in peanut-growing countries where the crops are produced under rain-fed conditions [7]. In the United States (US), the impact of aflatoxins could cost the peanut industry up to USD 58 million annually [8], making it an expensive problem for the agricultural industry. The US remains one of the largest producers of peanuts in the world, with more than USD 2 billion at the retail level and a farm value of over one billion dollars [9]. Lawley [10] reported that *A. flavus* is the most common fungus contaminating peanuts by producing carcinogenic aflatoxins, which destroy peanut shells before they are harvested. These bacteria also produce aflatoxins, which are both highly toxic and carcinogenic, thereby threatening humans and livestock. Pitt et al. [11] reported systemic infection of peanuts by *A. flavus* in soil and contaminated seeds, and Achar et al. [12] demonstrated, for the first time using electron microscopy, the seed-borne nature of *A. flavus* in Georgia peanuts and the establishment of the mycelium in seed tissues.

Despite the recognized impact and consequences of aflatoxin-producing fungi in peanut production, the measures in place to address this global issue remain limited [12,13]. Current strategies to control this fungus rely heavily on synthetic fungicides or preservatives belonging to the aromatic hydrocarbons, benzimidazoles [14]. Extensive use of these substances might produce several side effects, such as carcinogenicity, teratogenicity, and toxicity to consumers, as well as increased risk of high-level toxic residues in food products [15]. Chemical methods require sophisticated equipment and expensive chemicals or reagents [16].

The WHO recommends an integrated approach to control and prevent aflatoxin-affected crops at different stages of production (pre and post-harvest), policies and regulation on the levels of aflatoxin allowed, targeted farming practices, as well as seeking ways to remove the contamination [1]. A recent development in the field of biological management of aflatoxin in pre and post-harvested crops includes the successful application of competitive nontoxigenic strains of *A. flavus* and *A. parasiticus* [17]. The introduction of nontoxigenic strains of *Aspergillus* spp. in field studies of peanut and cotton has led to a significant reduction in aflatoxin contamination [17,18,19].

In addition, there is an increased interest in sourcing safer alternate natural products instead of synthetic chemical fungicides to combat *Aspergillus* spp. in the food chain. The use of essential oils with antifungal and anti-aflatoxigenic activity, which are environmentally friendly, generally regarded as safe (GRAS), and do not pose health risks, are currently being explored or used as biocontrol agents against this fungus. Several studies have reported antifungal properties of essential oils, with evidence of historical and long-term use of essential oils in human, health, and food settings for the prevention and management of fungal infections [14,16,20,21,22,23].

We investigated the antifungal activities of plant-based essential oils against *A. flavus* in Georgia peanuts. The peanut variety used in this study is the Tifguard, runner-type, which is highly recommended for peanut farmers by the United States Department of Agriculture [24]. It is the first peanut variety known to be resistant to two difficult pathogens; the peanut root-knot nematode (*Meloidogyne arenaria* (Neal) Chitwood race 1) and the tomato spotted wilt tospovirus [25]. In addition, we investigated the mode of action of selected essential oils on the morphological and ultrastructural changes in *A. flavus* and their impact on aflatoxin (AFB1) production. Hence, this study moves towards a novel, sustainable, eco-friendly solution to a serious problem of fungal contamination of peanuts and AFB1 production by the *A. flavus* in Georgia, USA and other peanut farming communities worldwide.

## 2. Materials and Methods

### 2.1. Isolation of A. flavus from Peanuts

Peanut seeds, variety Tifguard, runner-type, courtesy of Agricultural Research Service (ARS), Tifton, Georgia, US, were incubated on moist filter paper for seven days. *A. flavus* was isolated from a contaminated peanut and were directly plated onto potato dextrose agar (PDA) medium (Fisher Scientific, Waltham, MA, USA), and the plates were incubated with alternate periods of 12 h light and 12 h darkness for seven days. In addition, isolates from the contaminated peanut were compared to standard strains of *A. flavus* (ATCC 11498) from the American Type Culture Collection. The fungal colonies were observed under a light microscope (Leica, M13595, Leica Microsystems, Wetzlar, Germany) and were identified based on their macro and morphological characteristics such as the color of the colony, conidial heads, vesicle, phialides and conidia, using fungal keys and manuals [7,10,26,27,28,29]. Standard spore suspension of *A. flavus* was freshly prepared by the suspension of a loop full of spores from a 5-day-old pure culture plate in 5 mL of sterile water. The cell concentration of 10^−6^/mL in photo calorimeter was adjusted by diluting further with sterile water so that the optical density (OD) of the suspension was 0.01 at 460 nm. Yeast extract sucrose (YES) agar was used as a medium for aflatoxin production [30]. The isolated *A. flavus* was inoculated to YES agar medium, and the plates were sealed and incubated at a temperature of 27 °C in a CO_2_ incubator (Fisher Scientific, Isotemp, Waltham, MA, USA) for 10–15 days. After incubation, the plates were observed under ultraviolet light (UV) (Spectroline CC-80, Fisher Scientific, Waltham, MA, USA) to detect the presence of aflatoxin production. If the mold fluoresced under UV light, it was considered aflatoxin positive and confirmed as an aflatoxigenic form of *A. flavus*.

### 2.2. Selection of Plant-Based Essential Oils

Essentials oils (EOs) reported having antifungal properties against a myriad of *Aspergillus* spp. were selected. Fifteen essential oils were utilized in this study as follows: cedarwood (*Cedrus atlantica*), cumin (*Cuminum cyminum*), citronella (*Cymbopogon winterianus*), black pepper (*Piper nigrum*), cardamom (*Elettaria cardamomum*), cinnamon (*Cinnamomum verum*), ginger (*Zingiber officinale*), lemongrass (*Cymbopogon citratus*), orange (*Citrus sinensis*), spearmint (*Mentha spicata*), thyme (*Thymus vulgaris*), clove (*Syzygium aromaticum*), eucalyptus (*Eucalyptus globulus*), lavender (*Lavandula dentata*) and peppermint (*Mentha piperita*). All essential oils were purchased from Fisher Scientific and Sigma Aldrich, St. Louis, MO, USA. The essential oils were emulsified with 0.5% Tween 20 (*v*/*v*) stock concentrations of 10,000 ppm for further use.

### 2.3. Antifungal Activity Assay of Essential Oils

The in vitro antifungal activities of each of the essential oils (EO) from above, at different concentrations, were evaluated by the poisoned food technique [9]. Known volumes of EO and a commercial fungicide, prothioconazole (positive control), were incorporated into the potato dextrose agar (PDA) medium along with 0.5% of Tween 20 (*v*/*v*), which acts as an emulsifying agent to get the required concentrations of 125–4000 ppm. Further, the plates consisting of agar medium mixed with 0.5% of Tween 20 (*v*/*v*) without any essential oil was considered as blank control. A 5–10 µL conidial suspension from a 4–6-day-old *A. flavus* culture was inoculated in the center of the agar plates by using a capillary tube. The plates were incubated at 28 ± 2 °C, with an alternating period of 12 h of dark and light for 7 days until the mycelial growth in the control plates reached the edge of the plates. The efficacy of each EO as an antifungal agent was evaluated by measuring fungal colony diameter using a centimeter scale. Percentage inhibition of the radial growth with different oils compared to control was calculated using the following formula: Percentage mycelial inhibition (%) = [(*dc* − *dt*)/*dc*] × 100, where *dc* is the mean colony diameter for the control sets and *dt* is the mean colony diameter for the treatment sets.

### 2.4. Gas Chromatographic-Mass Spectrometry of Clove Oil

Based on the antifungal activity of all EOs from above, clove oil was selected and analyzed for its major composition using GC-MS (Shimadzu QP 2010 Plus, Tokyo, Japan), fitted with a flame ionization detector (FID) and Japan capillary column (0.32 mm i.d., length: 30 m, film thickness 0.25 µm). Injector temperature and ion source temperature were maintained at 280 and 230 °C, respectively. The oil sample (0.2 µL) was injected into the column with a split ratio of 80:1. The temperature program comprised 60 °C for 2 min, raised to 250 °C for 5 min at 10 °C/min and 280 °C for 15 min at 10 °C/min. The composition (%) was estimated with peak normalization and assuming its equal detector response for each run. The range of mass acquisition was 40–650 *m*/*z*. The peaks were detected by comparing the individual mass spectra with the reference database at the National Institute of Standards and Technology (NIST12 or NIST62) and Wiley 229 mass spectrometry libraries.

### 2.5. Transmission Electron Microscopy

Sporulating *A. flavus* mycelia treated with 500 ppm of clove EO were observed using transmission electron microscopy (TEM). Untreated mycelia served as the control. All samples were infiltrated with 1:1 then 1:2 ratios of ethanol to resin in a vacuum from four hours to overnight, then two changes of 100% resin under vacuum four hours to overnight. The samples were then fixed and allowed to evacuate overnight before being placed into an oven to polymerize for three to four days. The samples were trimmed and thin sectioned (~70 to 80 nm) using a diamond knife and RMC PT-XL Ultramicrotome (RMC Corporation, Tucson, AZ, USA). The sections were post stained with 7.5% uranyl acetate and Reynolds’s lead citrate. TEM micrographs of the samples were taken using JEM-1210 TEM instrument (JEOL USA Inc., Peabody, MA, USA) and operated at 90 kV. Ultrastructural alterations of the somatic and reproductive structures of treated and untreated samples were compared to assess the effect of clove EO against *A. flavus.*

### 2.6. Scanning Electron Microscopy

Sporulating *A. flavus* treated mycelia, with 500 ppm of clove EO, were also used for scanning electron microscopy (SEM) using standard chemical fixation and critical point drying methods. Untreated mycelia served as the control. Samples were fixed with 2.5% glutaraldehyde solution overnight at 4 °C. Thereafter, the samples were washed with 0.1 M sodium phosphate buffer solution (pH 7.2) three times for 20 min each. Following this, the samples were dehydrated in ascending ethanol series ending in three changes of 100% dry ethanol for about 45 min. Samples were dried in liquid carbon dioxide and were mounted on a silver stub and gold-covered by cathodic spraying (Polaron gold). The morphology of the fungus was observed using Topcon DS-130F Field Emission SEM/STEM (Topcon Technologies, Inc., Paramus, NJ, USA) and operated at 20 kV. Morphological alterations in the somatic and reproductive structures of treated and untreated samples were compared to assess the effect of clove EO against *A. flavus*.

### 2.7. Detection of Aflatoxin by Qualitative Methods

#### 2.7.1. Ammonia Vapor Method

ATCC 11498 is a toxigenic strain of *A. flavus*. The ammonia vapor (AV) method was used to confirm the aflatoxigenic form of *A. flavus* in contaminated peanuts, variety Tifguard, following protocol of Saito, et al. [31]. In addition, we determined the effect of clove EO on mycelial growth at different concentrations of the oil. Briefly, yeast extract sucrose (YES) agar plates were prepared by supplementing different concentrations of clove EO: 500, 1000, 1500, and 2000 ppm, respectively. YES plates without EO were treated as control. Ten microliters of the fungal spore suspension were inoculated at the center of the YES plates and were incubated at 25 ± 2 °C for five days. Following incubation, each dish was inverted, and approximately 200 µL of ammonium hydroxide solution (25%) was placed on the inside of the lid. The YES plates containing fungal mycelium and spores with different concentrations of clove EO were then inverted on the plates containing the ammonium hydroxide. EO-treated and untreated plates with active colonies were observed hourly for color change. A plum-red color change on the undersides of the plated colonies was an indication of aflatoxin producing strain of *A. flavus* isolate. No color change was categorized as a non-toxin producing strain. Mycelial growth was also monitored for plates impregnated with different concentrations of clove EO.

#### 2.7.2. Coconut Milk Agar Method

The coconut milk agar (CMA) method was used to further confirm the presence of aflatoxin in the aflatoxigenic form of *A. flavus* in contaminated peanuts, variety Tifguard, following the protocol of Davis et al. [32]. In addition, we determined the effect of clove EO on mycelial growth at different concentrations of the oil. CMA plates were prepared by supplementing with different concentrations of clove oil: 500, 1000, 1500, and 2000 ppm, respectively. CMA plates without oil were treated as control. Ten microliters of the fungal spore suspension were inoculated onto the solidified agar plates and were incubated at 25 ± 2 °C for five days. Following incubation, plates were exposed to 460 nm UV light and observed for fluorescence.

#### 2.7.3. Quantification of Aflatoxin by High-Performance Liquid Chromatography

Following incubation of clove EO-treated and untreated mycelia and spores of *A. flavus* on YES medium at 25 ± 2 °C for five days, extraction and purification of AFB1 were performed using High Performance Liquid Chromatography (HPLC; Waters 2790 HPLC-Photodiode Array UV detector, Milford, MA, USA) in accordance with standard protocols, using methanol as the solvent. Thirty grams of treated (500–2500 ppm) and untreated samples were collected from YES plates and homogenized with 30 mL of HPLC grade methanol. The mixture was vortexed for 5 min, followed by extraction and centrifugation. Extracts of 3 replicates were collected into a rotary evaporator flask, following which methanol was eliminated by evaporation under reduced pressure. The purified AFB1 from the treated samples was placed under a UV light along with AFB1 standard (Sigma Aldrich, St. Louis, MO, USA) that caused aflatoxin to fluoresce. Purified samples were stored at 4 °C in the dark for a couple of days prior to HPLC injection. A Waters 2790 Separations Module reversed- phase HPLC equipped with a photodiode array detector 2996 set between 200 and 500 nm was employed to capture the AFB1 spectrum in the sample. A Lunar C_18_ separation column, 100 mm by 4.6 mm internal diameter, with 5 µm packing material was used. A methanol/ultrapure water (60:40) mobile phase at a flow rate of 0.5 mL/min was used according to Vosough et al. [33] with minor modifications. The AFB1 standard equation was based on a four-point calibration curve and yielding the equation *Y* = 16976X + 8060.1, with a regression coefficient of *R*^2^ = 0.9957. The peak areas obtained from the fungal extracts for the control and clove EO treated samples were used in the regression equation to calculate the concentrations of AFB1.

### 2.8. Statistical Analysis

Data analysis was performed using statistical software SPSS for windows version 10.0.1 to calculate the means, standard errors, and standard deviations. One-way analysis of variance (ANOVA) was applied to the data to determine differences between treatments with significance levels set at *p* = 0.05. To check substantial differences between the levels of the mean factor, Tukey’s multiple comparison tests were applied to determine the levels of significance at 5% significance was applied.

## 3. Results

### 3.1. Isolation of A. flavus from Peanuts

After seven days of incubation, we observed healthy peanut seeds, variety Tifguard, germinating in a few plates (Figure 1a). Under light microscopic examination, these seeds showed the presence of both mycelia and spores of *A. flavus* (Figure 1b) on the cracked seed coat. After 10 days, seed germination was arrested, seed coats and radicles were completely covered with a mesh of mycelia and spores (Figure 1c). After 10–15 days of incubation, the *A. flavus* isolates on YES media fluoresced under UV light compared to the non-toxigenic forms.

### 3.2. Antifungal Activity Assay of Essential Oils

Fifteen essential oils (EOs) from above were tested for antifungal activities against the mycelial growth of *A. flavus* in peanuts. All EOs investigated demonstrated different levels of growth inhibition at different concentrations (Table 1). Total inhibition of mycelial growth was observed in 8 of 15 essential oils at concentrations between 500 and 4000 ppm. The most effective EO in inhibiting mycelial growth, based on the antifungal inhibition assay, was clove, with total inhibition at 500 ppm and over 90% inhibition at 250 ppm. This was closely followed by thyme, with total inhibition at 1000 ppm and approximately 90% inhibition at 500 ppm. EOs of lemongrass and cinnamon totally inhibited mycelial growth at concentrations ≤2000 ppm and at least 90% inhibition at 1000 ppm. The commercial fungicide, prothioconazole (positive control) completely inhibited the *A. flavus* at all tested concentrations ranging from 125 to 4000 ppm. As the concentration of most of the essential oils increased, the inhibitory effect also increased. Tukey’s multiple comparison test (*p* = 0.05) showed that all EOs tested, at different concentrations, showed varying degrees of antifungal activity against mycelial growth. Of all the EOs tested, clove EO was the most effective antifungal agent against this fungus, while ginger EO was the least effective, irrespective of the concentration.

### 3.3. Major Active Compounds of Clove Oil

Following the strength of the antifungal activities observed against *A. flavus* by clove EO from the diffusion assay, GC-MS was performed to investigate the major components of this oil. Our results show that the clove EO was predominantly composed of eugenol (83.25%), then E-caryophyllene (13.36%), followed by α-humulene (2.18%), which represented the marker compounds comprising approximately 99% of the whole oil (Table 2).

### 3.4. Effects of Clove Oil on Morphology and Ultrastructure of A. flavus Fungus

TEM analysis showed that in actively growing mycelia of *A. flavus*, untreated with clove EO, cellular components of conidia and hyphae remain unaltered. Untreated conidium showed a normal cell wall, and its cell matrix remained intact (Figure 2a). The plasma membrane had a linear shape, and the cell organelles, such as vacuoles, mitochondria, and nuclei appeared normal. The untreated mycelia had long strands of smooth hyphae with an intact cell wall. The hypha had a clear thick cell wall, and the cytoplasmic matrix remained unaltered (Figure 2b). In mycelia treated with clove EO (500 ppm), the main changes observed concerned the conidia and hypha. Conidia lost their normal shape and size, disintegrated cell surface, and lost their structural integrity in comparison to the control (Figure 2c). The treated hyphae suffered major alterations such as lysis of the cell wall, cytoplasmic disruption, the disintegration of mitochondria, and destruction of the nucleus. The hyphal wall appeared markedly thinner, and extensive vacuolization of the cytoplasmic matrix was clearly visible (Figure 2d). Treated hyphae also showed irregular septa (Figure 2e) and shrinkage of the cytoplasmic content (Figure 2f). Structural damage of mitochondria was prominent, and other cell organelles, including the nuclear content, were absent (Figure 2g).

SEM analysis showed that in actively growing mycelia of *A. flavus,* untreated with clove EO, the morphology of hyphae and conidia remained unaltered. Mycelium had long strands of linear, regular, rough-surfaced healthy hyphae (Figure 3a) with healthy conidiophores (Figure 3b) bearing healthy conidia (Figure 3c) and phialides (Figure 3d). Morphological changes were noticed in colonies incubated in media supplemented with clove EO. In these colonies, morphological alterations were prominent in both somatic and reproductive structures. The mycelia treated with clove EO (500 ppm) showed blistering of hypha (Figure 3e), with disrupted conidiophores bearing deformed vesicles (Figure 3f). In addition, in a few conidiophores, the chains of conidia were not visible and had fewer or destroyed phialides (Figure 3g). Affected phialides showed discrete damage or shrunken apex (Figure 3h). Overall, conidiophores showed a disrupted conidial chain, loss of conidia and collapsed vesicles in all treated mycelium (Figure 3i).

### 3.5. Detection of Fungus Aflatoxin

Using the AV method, we detected the aflatoxigenic form of *A. flavus* in peanuts, variety Tifguard. After exposure to ammonia vapor followed by one-hour incubation, the five-day-old mycelia and spores on YES media showed a plum-red colour change on the undersides of the plated colonies (Figure 4). This was an indication of aflatoxin producing strain of *A. flavus* isolate. YES plates, which served as the control, showed no colour change. We also observed that in plates impregnated with different concentrations of clove EO (500–2000 ppm), there was a significant reduction in mycelial growth, and no colour change was observed on the undersides of the plated colonies (Figure 4). In contrast, we observed a change in colour from pale white to dark pink in our EO untreated plates, indicative of an interaction between ammonia vapors and aflatoxin diffused into the YES medium.

Using the CMA method, we confirmed an aflatoxigenic strain of *A. flavus* in peanuts, variety Tifguard. We also observed that in plates impregnated with different concentrations (500–2000 ppm) of clove EO, the five-day-old mycelia and spores showed no blue fluorescence in the CMA plates (Figure 5). Furthermore, the 500 ppm concentration of clove EO appeared to be most effective in inhibiting aflatoxin production by this fungus. The blue fluorescence surrounding the colony was only detected in plates that served as a control. The absence of fluorescence was indicative of the action of clove EO on aflatoxin production by *A. flavus* in the CMA-treated plates.

### 3.6. Quantification of Aflatoxin (AFB1)

Qualitative analysis of the chromatograms in our study, indicated that the peaks (Figure 6A,B) appearing at retention times 5.23 min, confirmed the identification of AFB1 in the extracts, based on the AFB1 reference standard. Further, UV spectrum of AFB1 reference standard (Figure 6C), is identical to the UV spectrum (Figure 6D) of the clove EO-treated mycelia and spore samples of *A. flavus*, confirming the presence of AFB1 in the extracts. We also detected two unidentified peaks from the sample extracts. In addition, a plot of increasing clove EO versus AFB1 concentrations in sample extracts revealed a 10% decrease in the AFB1 concentration at 500 ppm clove EO compared to control, and to approximately 46% decrease at 2000 ppm clove EO, with a negative slope of −3.248 (ppm clove/ppm AFB1; Figure 7). Moreover, we established a correlation of 0.95, which is significant in terms of the effective suppression of AFB1 by clove EO. For quantitation of AFB1 to generate the results in Figure 7, the AFB1 calibration curve *Y* = 169758X + 80,464 with a regression coefficient of R^2^ = 0.9957 was based on the data provided (Appendix A). The calibration equation (Appendix A) obtained from Appendix A was used to calculate residual aflatoxin B1 based on the peak areas (Appendix A) of the clove oil treated samples. 

## 4. Discussion

Fungal contamination of peanuts is of utmost concern globally as peanuts are a rich and economical source of protein. The use of synthetic fungicides remains the common measure to reduce fungal contamination in the peanut industry in the US and elsewhere. It is widely known that these synthetic fungicides not only affect peanut consumers, but they can also be detrimental to the environment. Many strategies, including natural control, biological control, control of insect pests, development of resistant cultivar, have been studied for the management of contamination and aflatoxin production in crops. According to Reddy et al. [34], health hazards from exposure to toxic chemicals and economic considerations make natural plant extracts ideal alternatives to protect food and feed from fungal contamination. Plant-based antimicrobial compounds such as essential oils (EOs) and their bioactive compounds have been identified as potential biological control agents in the reduction/control of fungal contamination and toxin production, and more specifically as alternatives to synthetic fungicides against toxin-producing molds [35,36,37,38].

In our study, we screened fifteen essential oils at different concentrations against the mycelial growth of *A. flavus* in peanuts. The results demonstrate a dose-dependent inhibitory activity of all the EOs tested on the target fungus. Strong antifungal activity was observed in most of the EOs, demonstrated by the growth inhibition zones and minimal inhibitory concentration (MIC) values on the agar media. However, clove EO was the most effective of the EOs tested, against *A. flavus* mycelial growth and development, with a fungicidal effect at concentration 500 ppm. The commercial fungicide, prothioconazole, completely inhibited *A. flavus* at all tested concentrations ranging from 125 to 4000 ppm. In fact, the fungicide completely inhibited fungal growth at concentrations far below the reported ones. Similarly, previous studies have also demonstrated antifungal, inhibitory, and fungicidal action of different plants, including clove, thyme, cinnamon, lemongrass, tea tree, citronella, peppermint, and oregano oils [38,39,40,41,42,43,44]. Furthermore, earlier reports have demonstrated greater antifungal activity of thyme, lemongrass, and tea tree EOs against *Aspergillus* spp. compared to clove EO [42,45,46]. This is in contrast with our findings, as clove EO was more effective than all tested EOs and caused total inhibition of the mycelial growth. It is worth noting that our study focused on *A. flavus* in contaminated peanuts, which is different from several other studies that have investigated the antifungal activity of EOs against *Aspergillus* spp. in other hosts. To our knowledge, this is the first time that the antifungal activity of clove EOs has been demonstrated against *A. flavus* in contaminated peanuts, variety Tifguard, and our data pave the way to investigate other varieties of peanuts in Georgia.

It is widely known that the antimicrobial activity of essential oils is caused by various components that vary in their composition within the specific oil or other factors, which include plant type, seasonality, region, or location of where the plant is grown. Prindle and Wright [28] reported that the effect of essential oils (phenolic compounds) is concentration-dependent. At low concentrations, phenolic compounds affect enzyme activity, and at higher concentrations, they cause protein denaturation. Our GC-MS analysis showed that the clove EO was characterized by high amounts of eugenol (83.25%), with E-caryophyllene (13.36%) and α-humulene (2.18%) as marker compounds comprising approximately 99% of the whole oil. The predominantly high concentration of the eugenol is similar to that observed in the study by Pinto et al. [47]. In a recent study, the authors [44] found the IC_50_ of clove and cinnamon EOs to be >2000 µg/mL; however, the concentration of eugenol in the clove EO was determined to be 54% of the whole EO. This might explain the variations in the dose-responses between their study and ours, with a four-fold difference in the effective dose of clove EO (ppm) required for inhibition of mycelial growth of the *A. flavus*.

We further investigated the effects of the clove EO on mycelia and conidia of *A. flavus* by electron microscopy. A crucial observation of the ultrastructural changes, under TEM analysis, was cell wall lysis, thinning of the cell wall, cell membranes, and extreme vacuolization of the cytoplasmic matrix of hyphal cells. The most prominent observation was the damage to mitochondria and complete disintegration of other cell organelles, and most importantly, the nucleus was absent. Our observations corroborate the findings by Khosravi et al. [48]. These authors found changes in the cellular contents, including the cell wall, plasma membrane, and membranous organelles such as the nuclei and mitochondria of *A. fumigatus* and *A. flavus,* following exposure to the different EOs. According to [49], the antifungal activity of dill oil results from its ability to disrupt the permeability barrier of the plasma membrane and from the mitochondrial dysfunction-induced reactive oxygen species (ROS) accumulation in *A. flavus.* They further stated that mitochondrial damage leads to loss of ATP synthesis process in the cell, which is required for the fungi. We noticed excess vacuolization in our clove EO treated cells. It is reported that fungal vacuoles are responsible for the synthesis of many enzymes involved in glyoxylate pathways and fatty acid oxidation; thus, damage to vacuoles results in loss of these functions [50]. The review by Miri et al. [51] on the effect of essential oils on growth and aflatoxin production by *A. flavus* summarized that EOs could also coagulate the cytoplasm and damage lipids, proteins, cell walls and membranes that can lead to the leakage of macromolecules and the lysis. Data from our TEM analysis clearly showed that clove EO caused ultrastructural modifications of *A. flavus* hyphal cells leading to destruction and disintegration of the cellular matrix, and these changes may be attributed to a disruption in the enzymatic system of these cells by clove EO. Destruction of the cytoplasmic content might have certainly resulted in disruptions in metabolic activities of affected hyphal cells, ultimately affecting normal mycelial growth as observed in our study. In addition, since in our study, clove EO showed a negative impact on the conidial structural integrity, such changes certainly might have contributed to the mycelial growth of *A. flavus*.

The negative impact of clove EO on the mycelia and conidia of *A. flavus* was also observed in our SEM study. Our findings demonstrate that morphological alterations were prominent in both somatic and reproductive structures. Zambonelli et al. [52] observed inhibition of fungal growth, in addition to degeneration of fungal hyphae after treating them with *Thymus vulgaris*, *Lavandula* R.C. hybrid, and *Mentha piperita* essential oils. This type of damage to the hyphal morphology of plant pathogenic fungi following exposure to essential oils has also been reported by others [53,54] after exposure to EOs of the *Cymbopogon* species. Similar to our observations, the SEM study of *A. ochraceus* cells fumigated with natural cinnamaldehyde, citral, and eugenol showed alteration in the morphology of the hyphae, which appeared collapsed, with abnormal branching of hyphae in the apical region and loss of linearity [55].

Another key finding of our study was that growth inhibitions of the *A. flavus* were consistent with corresponding morphological changes of the hyphae and reproductive structures exposed to clove EO. The changes in hyphal morphology were certainly related to the loss of structural integrity of the cell wall supported by our TEM analysis, and further suggests that clove EO might have interfered with fungal wall synthesis directly or indirectly. It has been reported that several compounds of EOs, due to their lipophilic nature, cross fungal cell walls and membranes, interacting with membrane proteins and enzymes, disrupting cell metabolism, ultimately leading to cell death [56,57]. That clove EO, an antifungal agent, resulted in both morphological and ultrastructural alterations of *A. flavus* mycelium and spores is demonstrated for the first time by electron microscopy in our contaminated peanuts, variety Tifguard.

In addition, after confirming the presence of aflatoxin producing isolates of *A. flavus* in the contaminated peanuts by qualitative methods, we further established that clove EO also had an inhibitory effect on AFB1 production by this fungus. Similar to our studies, HPLC has been employed previously to detect aflatoxin production by *A. flavus* cultured on CMA and other differential media exposed to ammonia vapor pressure [58,59,60]. Other reports using such techniques demonstrated the potential of antifungal plant extracts against *A. flavus* growth and aflatoxin production in crops of economic importance [16,34,37,61,62,63,64]. Among these studies, some have found that clove oil inhibited the mycelial growth and aflatoxin production of *A. flavus* in rice [16,65]. The findings in our study support the conclusions of the previous research on the effect of clove EO on other crops, and at 500 ppm and above, a dose-dependent effect was observed, leading to a significant reduction in mycelial growth and AFB1 production. Hence, our study showed for the first time that clove EO showed both antifungal and antitoxigenic properties against *A. flavus* in the contaminated peanuts, variety Tifguard.

In the United States, the State of Georgia is recognized for approximately 50% of the national peanut supply, thus aiding the global recognition of the US as the third largest peanut producer [66]. Despite extensive research by the peanut industry in the US, the use of synthetic fungicides to combat contamination and aflatoxin production is very much in use on a large scale. The roles of essential oils as a biocontrol agent against *Aspergillus* spp. growth and toxin production in peanuts are largely unexplored, and our study offers new insight into this area. In addition, none of the earlier reports indicate that clove EOs have been tested for their fungi-toxic and antiaflatoxigenic potential against *A. flavus* in peanut growing states. Moreover, we are not aware of any published studies investigating the antifungal effect of essential oils or, in particular, clove against *A. flavus* in peanuts of the variety, Tifguard. Hence, we demonstrated for the first time the implication of clove EO on the deleterious morphological and ultrastructural alterations of somatic and reproductive structures of *A. flavus* and its ability to reduce AFB1 production by *A. flavus* in one variety of peanut. Our findings increase the possibility of exploiting the clove EO as an effective alternative to synthetic chemicals and as an effective bio-control and non-toxic bio-preservative to increase the quality and safety of peanuts varieties in the US and possibly around the globe. Most importantly, eugenol, the main compound in clove EO, can be further exploited to determine the EO mechanism of action and obtained data on its in vitro efficacy in both liquid and vapor form, and possibly finds its way as a lead compound in integrated pest management (IPM) to control *A. flavus* and to eliminate the carcinogenic aflatoxin B1 in peanuts.

## Figures and Tables

**Figure 1 jof-06-00383-f001:**
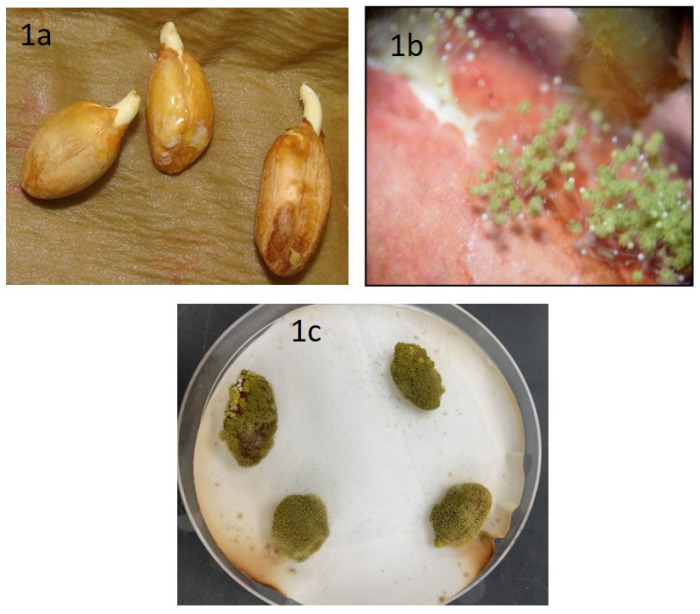
Isolation and screening of *A. flavus* from peanut seeds, variety Tifguard. Image of (**a**) healthy peanuts, (**b**) fungus sporulation from infected peanuts under light microscopy examination, and (**c**) peanuts infected with *A. flavus*, incubated for 7 days in moist filter paper.

**Figure 2 jof-06-00383-f002:**
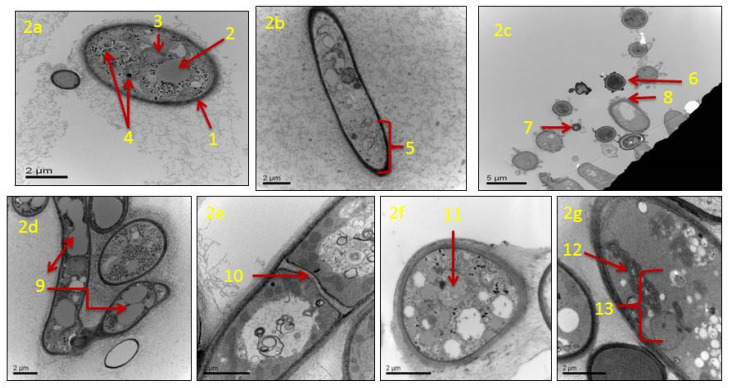
Transmission electron micrographs of *Aspergillus flavus* treated with 500 ppm of clove essential oil after seven days of incubation: (**a**) untreated healthy conidium with cell wall and cell-matrix intact; cell wall (*1*), vacuole (*2*), mitochondria (*3*), and nuclei (*4*); (**b**) untreated hypha with a clear thick cell wall and cytoplasmic matrix intact (*5*); (**c**) cross section of treated fungal conidia with an irregular shape (*6*), size (*7*) and disintegrated cell surface (*8*); (**d**) treated hyphae with extensive vacuolization in the cytoplasmic matrix (*9*); (**e**) treated hyphae with irregular septa (*10*); (**f**) treated hyphal cell with shrunken cytoplasmic contents (*11*); (**g**) treated hyphal cell with irregular/disrupted mitochondria (*12*), the absence of the nucleus and other cellular organelles (*13*).

**Figure 3 jof-06-00383-f003:**
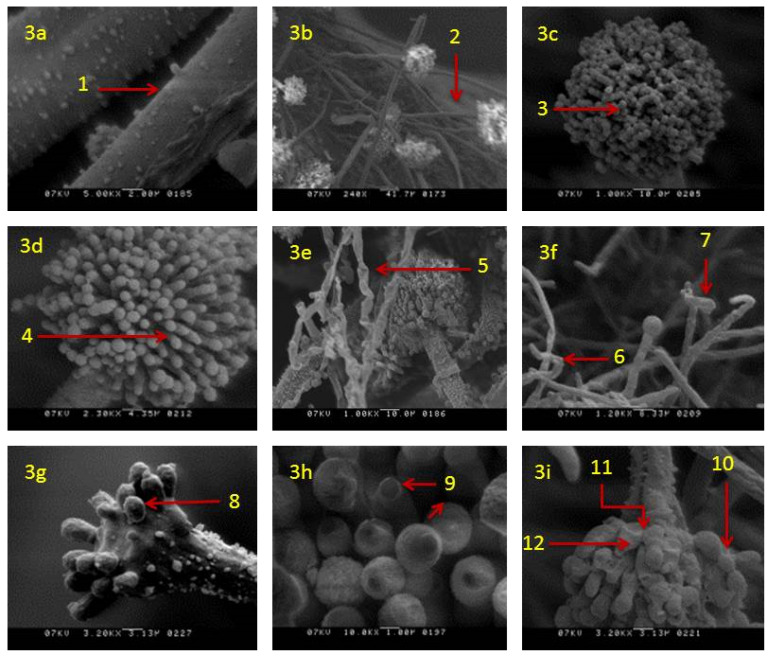
Scanning electron micrographs of *A. flavus* treated with 500 ppm of clove essential oil after seven days of incubation: (**a**) Healthy hyphae rough surface (*1*); (**b**) healthy hyphae with healthy conidiophore (*2*); (**c**,**d**) healthy conidium (*3*) and phialides (*4*); (**e**) treated mycelia with blistering hyphae (*5*); (**f**) treated hyphae with disrupted conidiophores (*6*) and deformed vesicles (*7*); (**g**) treated conidiophore with fewer or destroyed phialides on vesicle (*8*); (**h**) treated phialides with damage/shrunken apex (*9*); (**i**) treated conidiophores with disrupted conidial chain (*10*) collapsed vesicles (*11*) and phialides (*12*).

**Figure 4 jof-06-00383-f004:**
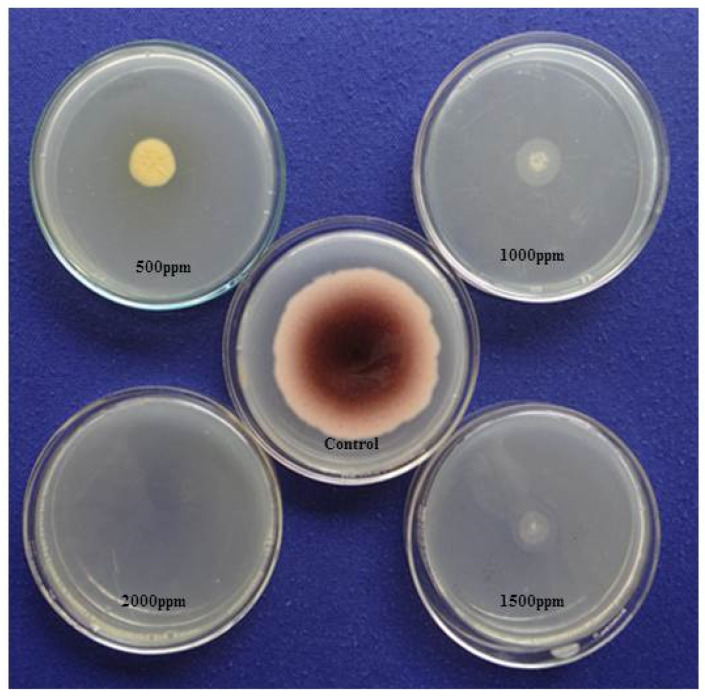
Effect of clove essential oil on aflatoxin (AFB1) production by *A. flavus* on yeast extract sucrose (YES) medium exposed to ammonia vapor.

**Figure 5 jof-06-00383-f005:**
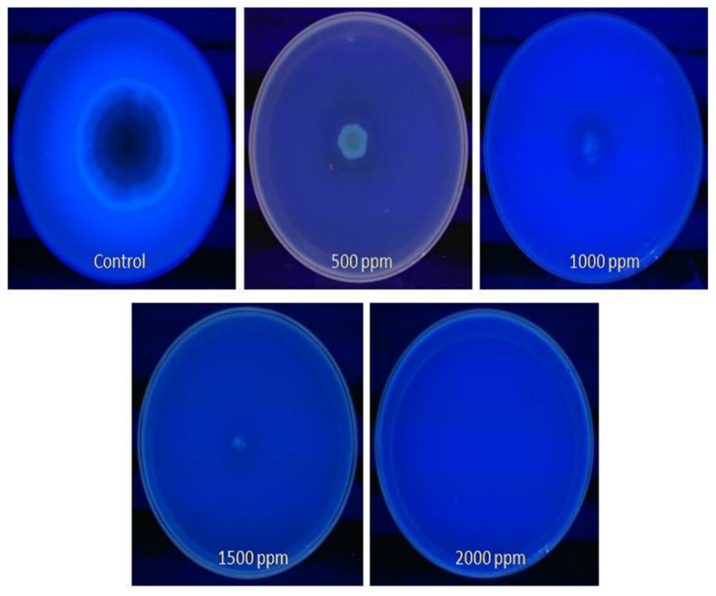
Detection of aflatoxin (AFB1) production by *A. flavus* treated with clove essential oil on CMA observed under UV radiations.

**Figure 6 jof-06-00383-f006:**
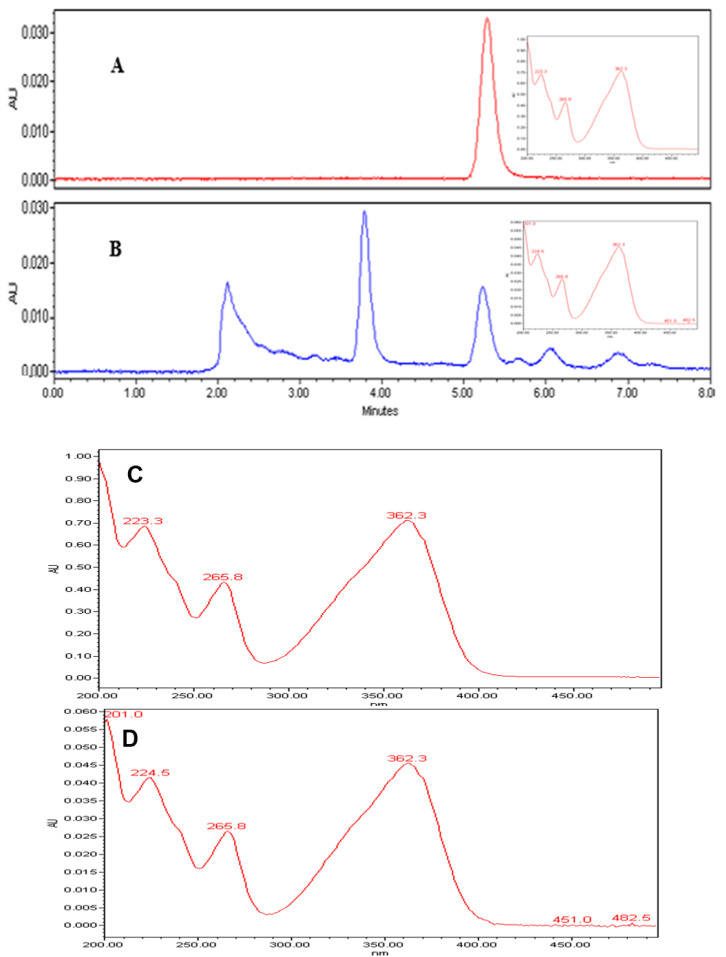
Chromatogram (**A**)—the peak of AFB1 standard at 40 ppm; chromatogram (**B**)—the peak of AFB1 from the extract at 500 ppm clove oil; the retention times of AFB1 average at 5.23 min in both chromatograms and additional peaks in chromatogram B were not identified in this study. UV spectra (**C**,**D**) are identical to the peaks eluting at 5.23 min in chromatograms A and B, confirming the presence of AFB1 in the extracts.

**Figure 7 jof-06-00383-f007:**
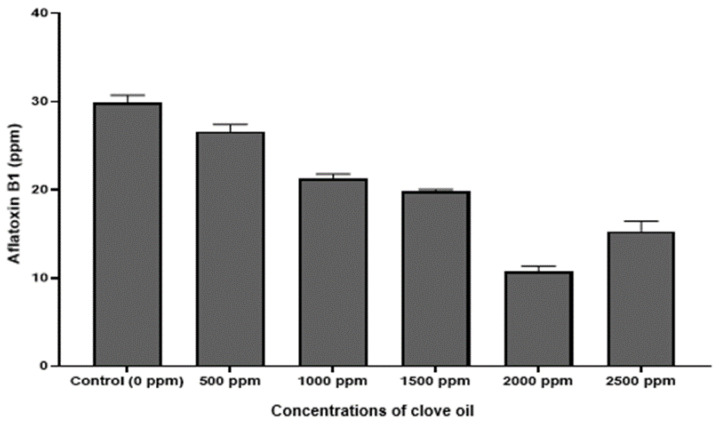
Column plot of the relation between the clove oil used versus residual aflatoxin (AFB1) after exposure of *A. flavus* mycelium to clove oil at different concentrations (ppm).

**Table 1 jof-06-00383-t001:** Inhibitory effect (%) of fifteen essential oils measured by the zone of clearance on the mycelial growth of *A. flavus,* isolated from peanuts, after 7 days of incubation.

Essential Oils	Percentage (%) of Mycelial Inhibition of *A. flavus* from Contaminated Peanuts Concentrations (ppm) of Essential Oils					
	125	250	500	1000	2000	4000
Clove	82.75 ± 2.12 ^a^	93.63 ± 0.00 ^a^	100 ^a^	100 ^a^	100 ^a^	100 ^a^
Thyme	74.62 ± 1.75 ^b^	79.12 ± 0.00 ^b^	88.87 ± 3.12 ^b^	100 ^a^	100 ^a^	100 ^a^
Lavender	15.25 ± 2.29 ^f^	29.62 ± 1.23 ^f^	48.75 ± 2.46 ^e^	51.25 ± 1.92 ^f^	64.37 ± 2.11 ^d^	88.00 ± 2.53
Cumin	15.62 ± 1.65 ^f^	28.37 ± 3.86 ^f^	43.75 ± 1.29 ^f^	51.62 ± 2.28 ^f^	65.87 ± 3.86 ^d^	91.25 ± 2.38 ^b^
Spearmint	24.37 ± 1.36 ^e^	33.75 ± 3.05 ^e^	48.62 ± 2.28 ^e^	58.37 ± 2.40 ^e^	85.32 ± 2.84 ^c^	100 ^a^
Cardamom	27.12 ± 2.78 ^d^	40.37 ± 2.75 ^d^	47.75 ± 3.62 ^e^	60.00 ± 2.65 ^e^	81.25 ± 3.59 ^c^	100 ^a^
Ginger	0.00 ^i^	0.00 ^i^	7.75 ± 1.11 ^i^	14.62 ± 1.14	36.12 ± 2.83	57.00 ± 2.50 ^d^
Lemongrass	33.37 ± 1.56 ^c^	56.25 ± 2.83 ^c^	84.87 ± 3.39 ^c^	92.75 ± 3.52 ^b^	100 ^a^	100 ^a^
Cinnamon	31.62 ± 3.21 ^c^	77.87 ± 3.42 ^b^	85.25 ± 2.74 ^bc^	91.00 ± 2.50 ^b^	100 ^a^	100 ^a^
Peppermint	21.25 ± 2.47 ^ef^	27.87 ± 1.76 ^fg^	29.62 ± 2.06 ^h^	48.37 ± 2.87 ^fg^	54.75 ± 3.47 ^f^	86.87 ± 1.73 ^c^
Citronella	25.50 ± 1.27 ^e^	31.50 ± 2.18 ^f^	53.75 ± 1.86 ^d^	85.62 ± 2.64 ^c^	94.00 ± 1.68 ^b^	100 ^a^
Black pepper	10.50 ± 1.11 ^g^	25.50 ± 2.33 ^g^	41.62 ± 2.39 ^f^	50.37 ± 3.71 ^f^	63.87 ± 1.28 ^d^	91.62 ± 2.89 ^b^
Cedarwood	16.62 ± 2.12 ^f^	24.50 ± 1.78 ^g^	31.25 ± 1.07 ^g^	45.87 ± 2.73 ^g^	60.87 ± 1.69 ^e^	94.12 ± 3.68 ^b^
Orange	2.37 ± 1.25 ^h^	17.50 ± 2.10 ^h^	27.25 ± 2.25 ^h^	33.37 ± 1.69 ^h^	59.62 ± 2.78 ^g^	84.62 ± 2.45 ^c^
Eucalyptus	23.50 ± 3.21 ^e^	35.37 ± 1.95 ^e^	48.87 ± 3.01 ^e^	73.37 ± 2.28 ^d^	82.62 ± 3.42 ^c^	100 ^a^

Mean ± standard deviation (*n* = 3). Values within each column followed by different letters are significantly different (*p* < 0.05). Note: A commercial fungicide (prothioconazole) was used as a positive control. The fungal colony without any treatment was considered as a blank control in which the growth of the *A. flavus* reached the edge of the plates after 7 days of incubation.

**Table 2 jof-06-00383-t002:** GC-MS analysis of clove essential oil showing major chemical components; RI: retention index literature comparison; the remaining compounds were in traces (<0.1%).

Clove Essential Oil		
Components	RI	Area (%)
Eugenol	1392	83.25
E-caryophyllene	1424	13.36
α-humulene	1579	2.18
α-cubebene	1349	0.34
Caryophyllene oxide	1587	0.27
Siporlepcenihene	1452	0.22

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
