# Peer review of "Investigation of the Antifungal and Anti-Aflatoxigenic Potential of Plant-Based Essential Oils against Aspergillus flavus in Peanuts"

_jof, 2020, doi:10.3390/jof6040383_

Round 1
Reviewer 1 Report
- The antifungal activity data of positive control should be provided.
- The antifungal activity and cytotoxicity of clove EO and eugenol should be provided. EC50 values is important for the intensity of activity.
- LC-MS (qualitative and quantitative) could increase the persuasiveness of data for the detection of aflatoxin. Specifically, standard curve line should be presented.
Author Response
Reviewer 1:
- The antifungal activity data of positive control should be provided.
Response: Thanks for the suggestion. The fungal colony without any treatment was considered as positive control. The same has been added in the material and methods and also a “Note: The fungal colony without any treatment was considered as positive control in which the growth of the A. flavus reached the edge of the plates after 7 days of incubation” is mentioned as footnote of table 1.
2.The antifungal activity and cytotoxicity of clove EO and eugenol should be provided. EC50 values is important for the intensity of activity.
Response: The authors would humbly like to state that antifungal activity of the clove oil has already been provided in the table 1. Since the focus of the current investigation was to screen various essential oils and not their bioactive compounds, authors did not analyze eugenol or any other chemical constituents of participating essential oils. Still, it would be interesting to compare and analyze various bioactive constituents of clove oil (eugenol, E-caryophyllene, α-humulene, etc.) as antimicrobials for our future studies and we thank the reviewer for his kind suggestion.
As far as cytotoxicity and EC50 analysis of clove oil is concerned, authors would like to submit that clove oil is an edible oil and comes under Generally Regarded As Safe (GRAS) category by the United States of Food Administration (1978, US-FDA, Washington DC). For Eugenol, cytotoxicity assay may be required but as the authors mentioned above, the current study was to demonstrate the antifungal property of clove oil per se and not the eugenol or any other biological constituents of clove oil, cytotoxicity test of clove oil was not performed. A further study in future may focus on cytotoxicity of clove EO and eugenol and EC50 values determined. We thank the reviewer for his kind suggestion.
- LC-MS (qualitative and quantitative) could increase the persuasiveness of data for the detection of aflatoxin. Specifically, standard curve line should be presented.
Response: The biochemical assay in the current study focused on quantification of aflatoxin B1 in our EO treated and untreated samples. A standard curve for quantification was obtained by plotting concentrations of the reference standard aflatoxin B1 versus the area counts from the HPLC/PDA chromatograms followed by linear regression analysis. Identification of aflatoxin B1 in the samples was straight-forward since the HPLC/PDA provided retention times as well as UV spectra that could be compared directly with the retention times and UV spectra of the pure reference standard. While mass spectrometry is clearly more powerful for molecular identification, we did not find a need to resort to LC-MS identification of aflatoxin B1. The standard curve line for aflatoxin B1 is being submitted under “supplementary data”. We thank the reviewer for his kind suggestion.
Reviewer 2 Report
Dear Colleagues!
I marked in yellow my point of view regarding Your manuscript. Verifies the writing of references and some corrections in the text, if you accept. Add for methods: Light microscopy.
Best regards.

Author Response
Reviewer 2:
I marked in yellow my point of view regarding Your manuscript. Verifies the writing of references and some corrections in the text, if you accept. Add for methods: Light microscopy.
Response: Remarks are highlighted in yellow in the PDF version (Enclosed). The authors have carefully attended to all remarks made by reviewer 2, highlighted in yellow and in addition, verified the writing in references and the running text.
We have not included “fifteen plant based essential oils” in the title of the manuscript as all the screened essential oils were not significant in controlling the aflatoxigenic A. flavus. Some of the oils have not shown any activities. Hence title of the manuscript retained as it is.
We added additional information of the light microscopy under methods line 96-97 and it reads: The fungal colonies were observed under a light microscope (Leica, M13595) and were identified based on their macro and morphological characteristics such as color of the colony, conidial heads, vesicle, phialides and conidia, using fungal keys and manuals. We thank the reviewer for his edit throughout the manuscript and his suggestion on the light microscopy.

Round 2
Reviewer 1 Report
The fungal colony without any treatment was the blank control. Commercial drug treatment group was considered as positive control. This is necessary. This not only reflects the activity level of essential oils, but also reflects the technical error during the experimental process.
Author Response
Dear Editor
This is with reference to our manuscript “Investigation of the antifungal and anti aflatoxigenic potential of plant-based essential oils against Aspergillus flavus in peanuts” being submitted to the special issue, Different Antimycotoxin Strategies, of the Journal of Fungi.
Answers to Reviewer 1 Round 2 comment
Reviewer 1:
The fungal colony without any treatment was the blank control. Commercial drug treatment group was considered as positive control. This is necessary. This not only reflects the activity level of essential oils, but also reflects the technical error during the experimental process.
Respond: At the outset, the authors apologize for wrongly mentioning the blank control as a positive control. Our control consisted of agar medium mixed with 0.5% of Tween 20 (v/v) without any essential oils. This procedure is inconsistent with the literature on essential oils to use a negative control to assess at screening level the antimicrobial activity especially when utilizing the agar incorporation or disk diffusion assay. We have now rectified the error in Sect 2.1 and table 1 including its footnote.
Regarding the positive control, we conducted an initial experiment with a demethylation inhibitor fungicide -Prothioconazole and compared it with the antifungal efficacy of our test samples. Prothioconazole completely inhibited the fungal strain at all tested concentrations ranging from 125-4000 ppm. In fact, the fungicide completely inhibited fungal growth at concentrations far below the reported ones. Hence, we the authors came to an agreement not to include the positive control in our study. Moreover, we all felt unreasonable to compare a pure and synthetic antifungal compound with natural essential oils (a combination of several compounds and all may not be antifungal) for an initial screening purpose. However, if the reviewer would like us to include the data of the positive control, we will be more than happy to insert it in Table 1.
We fully appreciate and thankful for the reviewer’s suggestions to strengthen our manuscripts.
Kind regards.